# Insights into Feruloylated Oligosaccharide Impact on Gel Properties of Oxidized Myofibrillar Proteins Based on the Changes in Their Spatial Structure

**DOI:** 10.3390/foods12061222

**Published:** 2023-03-13

**Authors:** Jingchao Yu, Shouwei Wang, Chengfeng Sun, Bing Zhao

**Affiliations:** 1China Meat Research Center, No. 70, Yangqiao, Majiapu East Road, Fengtai District, Beijing 100068, China; 2College of Life Sciences, Yantai University, No. 30, Qingquan Road, Laishan District, Yantai 264005, China; 3Beijing Academy of Food Sciences, No. 70, Yangqiao, Majiapu East Road, Fengtai District, Beijing 100068, China

**Keywords:** feruloylated oligosaccharides, myofibrillar proteins, oxidation, protein structure, gel properties

## Abstract

Polyphenolic compounds can protect against myofibrillar protein (MP) oxidation in meat products. In this study, the inhibitory effect of feruloyl oligosaccharides (FOs) on MP oxidation was investigated, and the gel properties of MPs were further studied. The results showed that 50–100 μmol/g protein of FOs could effectively inhibit damage to amino acid side chains by reducing carbonyl contents by 60.5% and increasing sulfhydryl and free amine contents by 89.5% and 66%, which may protect the secondary and tertiary structures of MPs. Additionally, FOs at 50 μmol/g protein had better effects on the crosslinking of MPs, leading to effective improvements in the gel properties, which can be seen in the rheology properties, scanning electron microscope (SEM) photographs, and the distribution of water in the MP gel. On the contrary, 150–200 μmol/g protein of FOs showed peroxidative effects on oxidatively stressed MPs, which were detrimental to MPs and contributed to their denaturation in the electrophoresis analysis and irregular aggregation in the SEM analysis. The concentration-dependent effects of FOs depended on MP-FOs interactions, indicating that an appropriate concentration of FOs has the potential to protect MPs from oxidation and enhance the gelation ability of pork meat during processing.

## 1. Introduction

Meat is the main source of high-quality protein and is widely accepted by consumers because of its good flavor and taste. As the main components of protein, myofibrillar proteins (MPs) play a crucial role in the gelation of meat proteins. They can form a stable gel network upon heating, giving meat products better sensory properties, texture, water-holding capacity, and oil retention [1]. The formation of MP gel is a complex thermodynamic process. With increasing temperature, the structure of MPs is denatured, and the spatial structure unfolds. Partially unfolded MP chains are interconnected and aggregate, finally forming a viscoelastic protein gel with a three-dimensional (3D) network structure [2]. The gel properties are affected by a variety of factors, such as pH, metal ions, and the oxidative induction of free radicals [3,4].

Meat is rich in unsaturated fatty acids, heme, metal catalysts, and other oxygen-promoting factors, which are easily oxidized and readily deteriorate during processing and storage. As preferential targets of free radicals, MPs show high oxidation reactivity, causing many problems in protein functionality and food quality. It was reported that the mechanism of protein oxidation resembled that of lipid oxidation, which is also a chain reaction of free radicals. The oxidation process also includes three stages: initiation, delivery, and termination. Free radicals attack the peptide and amino acid side chains, giving rise to protein fragmentation [5]. It was reported that oxidase, metal ions, and unsuitable processing environments, such as ion irradiation, can accelerate the production of oxidized substances, leading to the oxidation of proteins [6].

The excessive oxidation of meat products could lead to changes in the physical and chemical properties of MPs, including thiol loss, carbonyl compound formation, protein crosslinking, and amino acid modification, leading to the degradation of product quality [7]. These protein structure changes may lead to the degradation of protein functions, such as reducing the solubility, weakening the emulsification ability, and changing the viscosity characteristics. In processed meat products, protein oxidation has a direct impact on meat quality. Oxidation leads to changes in the physical and chemical properties of proteins, resulting in drip loss and changes in the texture and nutritional composition of meat products. Furthermore, protein aggregation during protein oxidation also leads to decreases in protein water retention and bioavailability. In addition, the carbonylation of oxidized proteins leads to the hydrolysis and off-color of meat products [8]. Not only are these modifications critical to the technical and sensory properties of meat products, but they may also have implications for human health and safety. Based on a recent report, the role of food protein oxidation in the loss of essential amino acids reduces digestibility and bioavailability, which leads to health disorders [9]. Therefore, it is of the utmost importance to understand the protein oxidation of meat products during processing to reduce the chances of humans ingesting potentially harmful compounds.

Given the above situation, various antioxidants have been widely used to prevent protein oxidation in a direct way. Plants are the most abundant source of antioxidants, attributed to their contents of various phenolic compounds, and play an important role in the meat industry. Clove extract (CE) and rosemary extract (RE) can significantly suppress protein oxidation and enhance the gel capacity of MP in dumplings stuffed with a meat-based filler [10]. Green tea extracts, such as catechins, are a source of nutraceutical supplements [11] that have been used as healthier natural antioxidants to replace carcinogenic antioxidants (such as butyl hydroxyanisole) and provide healthier products without changing their sensory acceptance. In addition, the antibacterial film prepared from ferulic acid has an antibacterial effect against *Escherichia coli* and *Staphylococcus aureus* and delays chemical changes in food ingredients [12].

Feruloylated oligosaccharides (FOs) are natural antioxidants with good water solubility and thermal stability and are mainly hydrolyzed from the dietary fiber in grains [13]. The structure of FOs consists of ferulic acid connected to a single arabinose residue through an ester bond, and the arabinose residue can be connected to a xylose residue through O-2 and O-3. FOs have good water solubility and heat resistance, which are not properties of many other natural antioxidants. They can be used as ingredients in functional foods to make health-care products. FOs contain both ferulic groups and hydrophilic oligosaccharides, making it possible to have the dual activity of ferulic acid and oligosaccharides. They provide superior protection against oxidative damage caused by free radicals [11]. It was found that the conjugated ferulic acid showed higher antioxidant activity than free ferulic acid, which was 48–58 times higher than expected [14]. FOs could also inhibit oxidative damage to red blood cells and DNA damage in human lymphocytes [13]. In conclusion, the good biological activity of FOs gives them good application prospects.

However, most of the current research on FOs focused on their antioxidation mechanisms in vivo and in vitro, and few have reported on their application in meat and meat products. Therefore, the purpose of this work was to explore the impact of different concentrations of FOs on the physical properties, chemical structure, and gel formation of oxidized myofibrillar proteins. In addition, the possible binding sites and conformational changes in MPs were further studied by Fourier transform infrared spectroscopy (FTIR), scanning electron microscopy (SEM), and low-field nuclear magnetic resonance (LF-NMR). The results would substantiate their protective effects on meat products as natural food antioxidants and provide theoretical support for the development of meat quality control technology.

## 2. Materials and Methods

### 2.1. Materials

The porcine loin was purchased from Wumart (Beijing, China), cut into small pieces, and then kept frozen. They were thawed in a refrigerator (4 °C) before use. FOs were extracted from wheat bran and purified. The purity of the reagents used in this experiment was analytical grade.

### 2.2. Preparation of Samples

#### 2.2.1. Extraction Process

MPs were extracted from the porcine loin according to the method of Park [15]. The thawed porcine loin was homogenized in a homogenizer (AM-3, Nihonseiki Kaisha Ltd., Nagaoka-shi, Japan) at 10,000× *g* for 1 min with phosphate buffer (1:5, *w*/*v*) containing 0.1 M NaCl, 2 mM MgCl_2_, 1 mM EDTA, and 10 mM Na_2_HPO_4_ (pH 7.0), followed by centrifugation (Thermo Sorvall LYNX 4000, Thermo Fisher Scientific Inc., Waltham, MA, USA) for 10 min at 10,000× *g* and 4 °C. The precipitate was preserved and washed 4 times with the same phosphate buffer as described above. After the final homogenization, the homogenate was passed through a four-layer gauze, and the pH was adjusted to 6.25. The filtrate was centrifuged at 15,000× *g* and 4 °C for 10 min. The MPs were stored at 4 °C and utilized within 48 h. Protein concentration was determined by the biuret method [16].

#### 2.2.2. Oxidative Treatments with Feruloylated Oligosaccharides (FOs)

MP solutions were diluted to the required concentration (20 mg/mL) with FOs at five final concentrations (0, 50, 100, 150, and 200 μmol/g protein). A hydroxyl radical (·OH) generation system (10 μM FeCl_3_, 100 μM ascorbic acid, and 1 mM H_2_O_2_) was used at 4 °C for 24 h to incubate. Samples were marked as nonoxidized MPs: NOX; oxidized MPs with 0, 50, 100, 150, and 200 µmol/g protein FOs: OX + 0, OX + 50, OX + 100, OX + 150, and OX + 200.

#### 2.2.3. Preparation of MP Gels

After oxidation, the samples were transferred to glass bottles (inner diameter × height = 1.65 cm × 5 cm) and then heated at 80 °C for 30 min. After heating, they were removed from the heat and cooled with ice water. Samples were stored in a refrigerator overnight.

### 2.3. Changes in MP Side Chains

#### 2.3.1. Carbonyls

The carbonyl content was measured using the 2,4-dinitrophenylhydrazine (DNPH) method. MP samples were mixed with a DNPH solution and precipitated with 20% trichloroacetic acid (TCA). After washing the recovered proteins to exhaustively remove unreacted DNPH, they were re-dissolved with 6 M guanidine hydrochloride (pH 2.3). The absorbance at 370 nm was read for the carbonyl content. A molar extinction coefficient of 22,000 M^−1^ cm^−1^ was used for the carbonyl content calculation.

#### 2.3.2. Total Sulfhydryls

The total sulfhydryl content was measured using Ellman’s reagent [17]. After dissolving them in a urea–sodium dodecyl sulfate (SDS) solution (8.0 M urea, 3% SDS, and 0.1 M phosphate buffer, pH 7.4), MP samples were incubated with 5,5’-dithiobis-(2-nitrobenzoic acid) (DTNB) reagent at 40 °C for 15 min. The absorbance at 412 nm was read, and a molar extinction coefficient of 13,600 M^−1^ cm^−1^ was used for the total sulfhydryl content calculation.

#### 2.3.3. Free Amines

The measurement of free amines was performed as described by Adler [18]. Briefly, MP samples were thoroughly mixed with 0.2 M phosphate buffer (containing 0.1% SDS, pH 8.2) and then reacted with 2,4,6-Trinitrobenzenesulfonic acid (TNBS) at 50 °C for 30 min in darkness. The reaction was terminated with 0.1 M Na_2_SO_3_. The absorbance at 420 nm was read, and a standard curve was prepared using L-leucine to determine the free amino content of the sample.

### 2.4. Changes in the Structure of MPs

#### 2.4.1. FTIR of MPs

The secondary structures of samples were measured using a Nicolet Is10 FTIR Fourier transform infrared spectrometer (Thermo Fisher Scientific, Waltham, MA, USA). Nonoxidized control and oxidized samples with different concentrations of FOs were diluted to 0.2 mg/mL protein concentration and scanned from 650 to 4000 nm. All spectra were averaged over three scans and corrected for the solvent signal. The data were processed by Fourier deconvolution to analyze the changes in the secondary structure of MPs.

#### 2.4.2. Surface Hydrophobicity

Surface hydrophobicity was determined using 1-anilino-8-naphthalenesulfonate (ANS) as a hydrophobic fluorescent probe [19]. The suspensions were diluted to 0.1, 0.2, 0.3, 0.4, and 0.5 mg/mL with Tris-HCl buffer containing 0.6 M NaCl (pH 7.0). An aliquot of 5 mL of the dilution was mixed by vortexing with 25 mL of Tris-HCl buffer containing 8 Mm ANS (pH 7.0). After 15 min in the dark, the fluorescence intensity of the ANS-complex protein was measured by a microplate reader (Synergy H4 BioTek Instrument Co., Ltd., Santa Clara, CA, USA). The excitation and emission wavelengths were 390 and 470 nm, respectively. Taking the protein concentration and the fluorescence intensity as the abscissa, the slope of the obtained curve is the hydrophobicity of the protein surface, which is represented by S_0_.

#### 2.4.3. Intrinsic Tryptophan Fluorescence

This was determined on an FS5 spectrofluorometer (Edinburgh Instruments Inc., Livingston, UK) with a dilute solution of MPs (0.1 mg/mL in phosphate buffer, 0.6 M NaCl, pH 6.25). The excitation wavelength was set at 280 nm, and the emission spectra were recorded from 300 to 450 nm at a 1 nm/s scanning speed. Data including both the samples and background were collected under the same conditions.

#### 2.4.4. Differential Scanning Calorimetry (DSC)

The thermal stability of MPs was analyzed by a differential scanning calorimeter (Q-2000, TA instrument, New Castle, DE, USA). First, 16~18 mg samples with a concentration of 40 mg/mL were accurately weighed and sealed in aluminum pans. A thermal scan was performed from 20 °C to 100 °C at a 5 °C/min heating rate. TA Instrument Explorer was used to analyze the maximum transition temperature (T_max_) and enthalpy changes (∆H).

### 2.5. Electrophoresis Analysis

Changes in the crosslinking and aggregation of MPs with oxidation and FOs were detected by polyacrylamide gel electrophoresis according to Laemmli’s method [20]. The concentrations of concentrated and separating gels were 4% and 12.5%, respectively. The relevant formations of MPs were observed using the Gel Doc XR+ with Image Lab Software (Bio-Rad Laboratories, Inc., Hercules, CA, USA).

### 2.6. Gel Strength and Water-Holding Capacity

The prepared gels were equilibrated at room temperature for 1 h before analysis. A TA-XT Plus texture analyzer equipped with a P/0.5 cylindrical probe was used to test the gel strength of the samples. The experimental parameters were as follows: test speed, 0.5 mm/s; trigger force, 5.0 g; and test distance, 10 mm.

The water-holding capacity (WHC) of the gels was determined from changes in the weight of the liquid. First, 5 g of MP gel was weighed in a tube and then centrifuged at 10,000× *g* for 15 min at 4 °C, and the liquid was then drained. The WHC of MP gels was calculated using the following formula:(1)$$\mathrm{WHC}\;\left(\%\right)=\frac{W_{2}-W_{0}}{W_{1}-W_{0}}\times 100$$
where *W*_0_ represents the weight of the centrifuge tube, and *W*_1_ and *W*_2_ represent the total weights of the centrifuge tube and MP gel before and after centrifugation, respectively.

### 2.7. Changes in the Dynamic Rheological Properties of MPs

The dynamic rheological properties of samples with a concentration of 20 mg/mL were measured using a DHR-2 rheometer (TA Instruments, New Castle, DE, USA) with 2 parallel plates (40 mm upper plate diameter, 1 mm gap). Gelation was induced by heating from 20 to 95 °C at a rate of 1 °C/min. During heating, the storage modulus (G’) and the loss modulus (G”) were constantly recorded to describe the dynamic rheological properties of samples.

### 2.8. Low-Field Nuclear Magnetic Resonance (LFMR) of MP Gel

The samples were placed into cylindrical glass tubes (10 mm in diameter) after heating them at 80 °C for 20 min. The transverse relaxation time (T_2_) was measured on an NMI20-040H-1 NMR analyzer (Niumag Analytical Instrumental Co., Suzhou, China) using the Carr–Purcell–Meiboom–Gill (CPMG) sequence at 22.6 MHz, with 32 scans, 10,000 echoes, 6.5 s between scans, and 200 μs between pulses of 90° and 180°. The low-field NMR relaxation times were analyzed with the MultiExp Inv Analysis software (Niumag Analytical Instrumental Co., Suzhou, China).

### 2.9. Scanning Electron Microscopy (SEM)

The gels were cut into small squares of 1 cm^3^ (1 cm × 1 cm × 1 cm) and fixed in 0.1 M phosphate buffer (pH 7.2) containing 2.5% glutaraldehyde for 24 h at 4 °C. After washing them with 0.1 M phosphate buffer (pH 7.2) three times, dehydration was performed using a series of ethanol (30, 50, 70, 80,90, and 100%, in this order) for 10 min. Samples were further dehydrated by freeze-drying, then sputter-coated with 10 nm of gold, and observed under 10,000 magnification through a Hitachi S-8020 SEM (Tokyo, Japan) operating at a voltage of 20 kV.

### 2.10. Determination of Chemical Force of MP Gels

The chemical force of MP gels was measured according to Yang’s method [21] with a slight modification. Chopped gels (1 g) were added to the reagents (10 mL) to break specific bonds: 0.05 M NaCl (SA), 0.6 M NaCl (SB), 0.6 M NaCl + 1.5 M urea (SC), 0.6 M NaCl + 8 M urea (SD), and 0.6 M NaCl + 8 M urea + 2% mercaptoethanol (SE); the content of ionic bonds is the difference between protein contents soluble in SB and SA, the content of hydrogen bonds is the difference between protein contents soluble in SC and SB, the content of hydrophobic interactions is the difference between protein contents soluble in SD and SC, and the content of disulfide bonds is the difference between protein contents soluble in SE and SD. Then, the mixtures were high-speed homogenized (10,000 r/min, 1 min) and stirred at 4 °C for 2 h. After centrifugation (10,000× *g*, 10 min, 4 °C), the protein concentration of the supernatant was determined.

### 2.11. Statistical Analysis

Experimental data were analyzed using ANOVA with SPSS 18.0 and tested using Duncan’s method with a 95% confidence level (*p* < 0.05). Results are expressed as the mean ± standard deviation.

## 3. Results and Discussion

### 3.1. Modifications of Amino Acid Side-Chain Groups

#### 3.1.1. Carbonyl Content

The side-chain amino acid functional groups of proteins are easily oxidized and readily form carbonyl derivatives, so the carbonyl content is widely regarded as one of the indexes to judge the extent of protein oxidation. The protein carbonyl generation pathway also includes the oxidation of amino acid side chains, the breaking of peptide bonds, and the introduction of exogenous carbonyl groups. For example, aldehydes in the oxidation products of fats and semiquinones or quinones in the oxidation products of phenols such as ascorbic acid are covalently linked to the protein side chain. Substances with phenolic hydroxyl groups can both promote and inhibit MP carbonyl formation, depending on the concentration of the phenol, oxidation conditions, and protein targets.

FOs showed powerful antioxidant activity and good water solubility, because they have both phenolic hydroxyl groups and oligosaccharide groups. The effects of FOs on the carbonyl content of MPs are shown in Table 1. The carbonyl content in the OX + 0 group was significantly increased (*p* < 0.05), reaching twice that in the unoxidized group. As the FOs were added, the carbonyl content of samples treated with 100 µmol/g FOs was only 2.34 μmol/g MP and decreased by 60.5% compared with the OX + 0 group. The results indicated that the FOs decreased the carbonyl content of MPs and effectively inhibited MP oxidation. The hydroxyl group on the b-ring of FOs is regarded as the scavenger of free radicals, and the dihydroxyl position is thought to be the metal-chelating site, so it also plays a key role in preventing the oxidation of ions from the side chains and the formation of carbonyls [22]. These results are consistent with previous observations that chlorogenic acid and tea polyphenols containing the dihydroxy part of the benzene ring generally blocked the formation of carbonyls in MPs [8,23]. Nonetheless, there was an upward trend in the carbonyl content with high doses of FOs (150 and 200 μmol/g protein). The reason for this could be that excessive FOs could react with MPs through covalent bonding or attraction between positive and negative charges, which leads to an increase in carbonyl content. However, the increased carbonyl content caused by FOs did not represent the intensification of the MP oxidation degree. It is similar to phenolic compounds in that different concentrations could also change the carbonyl content but were considered to exhibit both antioxidant and pro-oxidant activities [24].

#### 3.1.2. Total Sulfhydryls

The sulfhydryl groups in myofibrillar proteins are easily attacked by hydroxyl radicals and converted into disulfide bonds, resulting in protein crosslinking polymerization. Therefore, the reduction in sulfhydryl content is one of the important early indicators of protein oxidation caused by free radicals. The effects of FOs on the total sulfhydryl content of MP are shown in Table 1. The total sulfhydryl content in the OX + 0 group was significantly decreased (*p* < 0.05), reaching 54.31% that in the unoxidized group. The addition of 50 and 100 µmol/g protein FOs effectively inhibited the reduction in the total sulfhydryl content. However, 150 or 200 μmol/g of FOs did not show a protective effect. Compared with NOX, the sulfhydryl content of OX + 200 was lessened by 7.0%. It has also been observed that a high dose of phenolic acid could cause the reduction of sulfhydryl groups in MPs that were oxidatively stressed with chlorogenic acid (CA) [25] and rosmarinic acid (Ros-A) [26]. Studies have shown that polyphenols contain phenolic hydroxyl groups, which can easily be converted into quinones under oxidative conditions and then combine with proteins to form thiol–quinone compounds with nucleophilic groups [27]. This change does not mean the aggravation of the oxidation degree of MPs, but it also has an impact on the structure of MPs. Therefore, a low dose of FOs can be used to protect sulfhydryl groups, while a high dose of FOs may cause an excessive change in the MP structure and destroy the stability of MPs.

#### 3.1.3. Free Amine Content

The ε-NH_2_ group of lysine is very vulnerable to free radical attack and undergoes deamination reactions to transform into carbonyl groups, which are then covalently combined with −NH_2_ to further reduce the content of free amino groups. Consequently, the formed carbonyl groups may covalently combine with ε-NH_2_ to further minimize the content of free amines. The effects of FOs on the free amine content of MPs are shown in Table 1. The OX + 0 group was characterized by 40.1% lower free amine content compared to the NOX group due to amines being attacked by ·OH groups (*p* < 0.05). The addition of 50 or 100 μmol/g FOs prevented the oxidation of ε-NH_2_ groups and increased the free amine content, whereas high dose of FOs significantly decreased the content of ε-NH_2_ groups. In the study by Wang et al., the addition of high doses of rosmarinic acid significantly reduced the ε-NH_2_ group content of oxidized MPs compared with unoxidized MPs, similar to the results of our present study [27]. This may be due to the irreversible reaction between the quinone of FOs and the free amines of MPs under oxidative stress, in turn forming amine–quinone adducts. Utrera has also stated that rosmarinic acid will generate quinone due to oxidation and covalently bond with the ε-NH_2_ group in MPs, which additionally reduces the ε-NH_2_ group [28].

### 3.2. Changes in the Structure of MPs

#### 3.2.1. Secondary Structure of MPs

The changes in the MP secondary structure were analyzed by Fourier infrared spectroscopy [29]. The relative contents of MP secondary structures are shown in Table 2. It was observed that the β-sheet structure accounted for the largest proportion of secondary structures, which is the main structure required to maintain the MP gel system. Upon MP modification with H_2_O_2_, the relative content of α-helices underwent significant attenuation, denoting obvious losses of the α-helix structure of light myosin. The content of α-helices of OX + 50 was observed to significantly increase compared to the OX group, which may be ascribed to the antioxidant effect of the phenolic hydroxyl group in FOs. However, the α-helix fraction of MPs declined with increasing FOs concentrations (OX + 150 and OX + 200), which indicates a dose-dependent effect due to the conjugation of FOs. FOs may be oxidized to destroy the hydrogen bond formed between the carbonyl oxygen (C=O) and the amino hydrogen (−NH) in the proteins, thereby reducing the α-helix content. Relevant studies have pointed out that the combination of polyphenols would decrease the α-helix content of the protein, including salt-soluble MP [30] and lactoferrin [31]. The above results indicate that adding a low concentration of FOs can stabilize the secondary structure of MPs, while high doses of FOs may destroy the initial MP structure. Furthermore, another study exploring the effects of phenolic compounds on the secondary structures of proteins has reported conflicting results. It proved that epigallocatechin gallate (EGCG) may break the α-helix and transform it into a random coil, which increases the flexibility of the protein. On the contrary, catechin and gallic acid (GA) do not destroy the α-helix of the protein [32]. These differences may be due to the structural flexibility of proteins and polyphenols, as well as the concentration effect [33].

#### 3.2.2. Surface Hydrophobicity

The surface hydrophobicity of proteins is an important index to evaluate the functional and conformational changes in proteins, which can reflect the distribution of hydrophobic groups on the surfaces of proteins and plays an important role in stabilizing the tertiary structures of proteins and controlling the distribution of hydrophobic groups in protein molecules. The effects of FOs on the surface hydrophobicity of MPs are shown in Figure 1a. The oxidation treatment significantly promoted the unfolding of the internal structure of MPs, increasing the surface hydrophobicity. Additionally, the secondary structure of MPs changed from an α-helix to a β-sheet and β-turn, which was also a manifestation of the enhanced surface hydrophobicity. The FO addition (50 µmol/g protein) significantly reduced the surface hydrophobicity of MPs compared to oxidized MPs. Studies have shown that low-concentration polyphenols can bind to the pyrrolidine ring exposed by the proline residues of the protein through hydrophobic interactions, thereby reducing the surface hydrophobicity [34]. Furthermore, as the FO concentration increased, the surface hydrophobicity gradually diminished. Cao et al. [35] similarly reported a decrease in the protein surface hydrophobicity of MPs with the addition of phenolic compounds. In most instances, the binding of polyphenols to protein chains is known to cause partial unfolding or denaturation, resulting in possible changes in protein folding and protein functionality [36].

#### 3.2.3. Intrinsic Tryptophan Fluorescence

Protein endogenous tryptophan fluorescence is very sensitive to the polarity of its surrounding microenvironment, and the intensity of endogenous tryptophan fluorescence represents the folded or unfolded state of the protein. The effects of FOs on the intrinsic tryptophan fluorescence of MPs are shown in Figure 1b. The tryptophan fluorescence intensity of oxidized MPs was much higher than that of unoxidized MPs (*p* < 0.05), indicating that the protein structure was expanded, and the hydrophobic groups were exposed after oxidation. The fluorescence intensity of tryptophan showed a decreasing trend as the amount of FOs increased, showing a typical dose–response relationship, which indicated that tryptophan residues entered a more hydrophilic environment through the binding of FOs. The intrinsic tryptophan fluorescence of myofibrillar proteins was positively correlated with the hydrophobicity of the environment. It was also observed that the fluorescence intensity of tryptophan was significantly lower than that of oxidized MP when the addition of FOs was increased to 150 μmol/g and 200 μmol/g. The results showed that the introduction of excessive FOs not only promoted the structure expansion of MPs but also improved the polarity of the tryptophan environment, resulting in a decrease in fluorescence intensity. Similar changes were observed in the interaction of caffeic acid and quercetin with pork MPs, with the authors explaining that water-soluble phenolic acids could enhance the polar environment and produce a shielding effect, thereby reducing the overall fluorescence of MPs [37]. Simultaneously, excess phenolic hydroxyl and carboxyl groups of FOs might provide the basis for the increase in the system’s hydrophilicity.

#### 3.2.4. Thermal Analysis

DSC (Differential Scanning Calorimetry) is generally considered to be an important parameter for evaluating the thermal stability of protein structures during heat treatment. The effects of FOs on the thermal stability of MPs are shown in Table 3. Nonoxidized MPs showed three clear endothermic peaks with T_max_ values of 48.98, 56.80, and 65.31 °C, which are ascribed to the myosin head, myosin tail, and actin, respectively [38]. Oxidative stress shifted all three transition peaks and the corresponding T_max_ to lower values, which led to an unstable protein structure. The addition of 50 µmol/g FOs could protect the protein structure of MPs to a certain extent, and total enthalpy (ΔH) was also increased (*p* < 0.05). ΔH is the net value of the combination of endothermic changes (breaking of hydrogen bonds) and exothermic changes (hydrophobic interactions or protein aggregation) [39]. The increase in ΔH indicated that hydrophobic interactions or protein aggregation was reduced, and the protein structure remained stable. It is inferred that polyphenols could be oxidized to quinones or semiquinones, which can covalently react with nucleophilic amino acids such as cysteine or lysine on proteins to protect MPs from oxidation [40]. Identical to the result of tryptophan fluorescence, the thermal stability of the myosin tail in the oxidized MP modified with 200 μmol/g FOs was the lowest (ΔH_1_: 0.0083). In addition, the denaturation peaks of the myosin head and actin disappeared, which indicated that the structure of the oxidized MPs was affected by the interaction between FOs and MP. Furthermore, the addition of high concentrations of FOs may destroy the structure of MPs, resulting in a decrease in ΔH. In brief, the addition of a high dose of FOs might promote MP structure destabilization, resulting in a decrease in the total enthalpy. This result is consistent with the findings from the above α-helix reduction and tryptophan fluorescence results.

### 3.3. Crosslinking of MPs

Figure 2 shows the effect of FOs on crosslinked MP aggregation. Under nonreducing conditions, the myosin heavy chains (MHCs) of oxidized MPs were significantly reduced compared with nonoxidized MPs, and some polymeric aggregates appeared at the top of the resolving gel, indicating that protein crosslinking and aggregation occurred in oxidized MPs without the addition of FOs. In contrast, under reducing conditions, most of the MHCs were recovered, reflecting that the protein aggregation caused by oxidation was mainly attributed to the crosslinking of MHCs through disulfide bonds [41]. It is worth noting that the actin in the oxidized MP samples did not show many changes, despite major structural damage. This may be attributed to the absence of free SH in actin molecules [42]. According to Figure 2b, we found that the MHC and actin bands in the OX, OX + 50, and OX + 100 groups were close to those in the NOX group, while the recovery bands in OX + 150 and OX + 200 were weaker. This result showed that the moderate addition of FOs had a certain antioxidant effect, which hindered protein polymerization, but the high content of FOs may further react with MPs in the oxidation system. Tang [38] proved that oxidized plant polyphenols could generate quinones and promote the formation of disulfide bonds as well as be used as a crosslinking agent to produce protein aggregates through their addition to free nucleophilic protein side-chain groups, explaining the reduction in free amine and sulfhydryl contents.

### 3.4. Gel Properties of MPs

#### 3.4.1. Rheological Properties

The elastic storage modulus (G’) and loss modulus (G”) are important parameters of the dynamic rheological properties of protein gels and can reflect the elasticity and viscosity of protein gels, respectively. The effects of FOs on the dynamic rheological properties of pork MPs are shown in Figure 3. The nonoxidized MPs displayed a typical G’ pattern with a rheological transition peak occurring at about 48 °C and 65 °C, given their respective denaturation temperatures revealed by the DSC results. This could be due to the degeneration of the myosin head and uncoiled myosin tails, which mean a disruption of the protein network to form gel network structures. Nevertheless, the development of G’ varied between sample treatments. Oxidized MPs exhibited a slight G’ decrease in both transitions but a remarkable reduction at the end of the heating process, indicating that both myosin head–head and tail–tail interactions were impaired. However, the presence of FOs at 50 µmol/g increased both the scope of the rheological transitions and the final G’ value, indicating that a moderate dose of FOs was conducive to the interaction and aggregation of the above proteins during the heating process. Consistent with this result, FOs had a positive effect on the content of α-helices and thermal stability compared to those of oxidized MPs but induced further losses of surface hydrophobicity and tryptophan fluorescence. The further enhancement of gelation induced by the appropriate concentration of FOs may be mainly attributed to the further aggregation of MPs, allowing more functional groups to participate in the gelation process, such as the disulfide bond. The addition of FOs at 200 µmol/g, however, altered the gelation behavior of MPs: the two typical transition peaks disappeared, and the G’ value sharply decreased, indicating that immoderate FOs could extensively destroy the structure and cause the excessive aggregation of MPs. Similarly, Cao and Xiong [23] found that a high concentration (150 µmol/g) of chlorogenic acid was detrimental to MP gelation.

The value of G” represents the viscosity characteristics of the gel system. The changes in this value showed a pattern similar to that of the first peak of the G’ value when the temperature reached 55 °C. The G” value dropped sharply from 55 °C to 62 °C and then continued to increase before the temperature reached 85 °C. During the heat treatment, the G’ of the samples was always higher than G”, indicating that the viscous property was dominant during the gel formation process. Continuous molecular crosslinking and interactions generated during heating make the elastic properties dominant in the gel system, resulting in the improved gel strength of the mixed system. The fact that higher G” values were obtained with 50 µmol/g FOs but the lowest G” values were obtained with 200 µmol/g FOs revealed that a moderate concentration of FOs interacted with MPs to produce a gel network with viscoelastic properties in the continuous phase. Cao and True [42] pointed out that superfluous gallic acid impaired the MP gelling potential by hindering the formation of an elastic protein network and, therefore, gave rise to the poor water binding of the gel, which can explain the results of our study.

#### 3.4.2. Gel Strength and Water-Holding Capacity of MPs

Gel strength can reflect the ability of myofibrillar proteins to form a gel, which is closely related to the gel structure. As shown in Figure 4, the gel strength of MP gel was significantly increased when it was oxidized (*p* < 0.05). The gel strength tended to gradually increase with the concentration of FOs, which indicates that FOs can improve the performance of MP gels by preventing damage by hydroxyl radicals, resulting in the continual increase in the intensity of gels. However, high concentrations of FOs (≥100 μmol/g) significantly disrupted the strength of the gels (*p* < 0.05), suggesting that the presence of higher doses of FOs would lead to the severe crosslinking and aggregation of MPs, thus leading to an undesirable gel network [43].

In the processing of meat products, the water retention of gel is also a crucial property that is related to the tenderness, juiciness, and taste of meat products. As shown in Figure 4, 1 mM hydroxyl radicals increased the binding of water in the MP gel system. After adding 50 μmol/g FOs, the water-holding capacity increased by 24.1% compared with unoxidized MPs (*p* < 0.05), owing to FOs’ ability to promote the expansion of the protein structure and improve the hydrophobic interactions that maintain the gel structure. However, with the increase in FO concentration, the water retention capacity of the gel decreased significantly (*p* < 0.05). This may be due to the poor gel network formation caused by excess FOs, or even the destruction of the three-dimensional network structure, resulting in a low amount of water remaining in the gel matrix. Tang et al. [30] found that when they added 1.25 mM rosmarinic acid to oxidized MPs, the cysteine in MPs would form a complex with rosmarinic acid, which would reduce the formation of disulfide bonds, resulting in weakened gel strength and poor water retention. Warner reported that poor water-holding capacity (WHC) compromised the visual acceptability, weight loss, cooking yield, and organoleptic properties of meat and meat products at the point of consumption [44].

#### 3.4.3. Distribution of Water in MP Gel

Low-field NMR can be used to analyze the water distribution and migration in the oxidized MP gel system mediated by FOs through T_2_ relaxation time, which can indirectly reflect the gel quality of MPs [45]. The analysis is divided into three peaks, which are T_2b_ (0–10 ms), T_21_ and T_22_ (100–1000 ms), and T_23_ (1000–2000). Peak 1 (T_2b_) represents the water binding with hydrophilic groups, peak 2 (T_21_ and T_22_) represents the water trapped in the three-dimensional structure of MPs, and peak 3 (T_23_) represents free water. Table 4 gives the relaxation times (T_2b_, T_21_, T_22_, and T_23_) and the corresponding proportions (P_2b_, P_21_, P_22_, and P_23_) of MPs. It can be seen from Table 4 that compared with unoxidized MPs, the water bound to OX-MPs did not change significantly (*p* > 0.05), but the nonflowing water was significantly reduced (*p* < 0.05), whereas free water was significantly increased (*p* < 0.05), indicating that oxidation treatment could reduce the binding of water in the gel system and enhance the degree of freedom. When 50–100 μmol/g FOs were added to the sample, the nonflowing water in the gel was significantly reduced (*p* < 0.05). The study by Gravelle [46] indicated that the T_2_ relaxation time reflected the degree of interaction between the substance and its chemical environment, where a shorter relaxation time indicated a faster exchange process and stronger interactions. The results showed that the existence of a low concentration of FOs stabilized the water distribution of the gel system and reduced the mobility of the water. However, when the amount of FOs reached 200 μmol/g, the relaxation times of T_21_ and T_22_ in the mixed gel increased significantly (*p* < 0.05), which may be due to the formation of protein–quinone structures between excess FOs and MPs, resulting in the congestion of protein and water.

The corresponding proportions of peaks can represent the proportions of different states of water under different treatments, so as to infer the migration of water [47]. It can be seen from Table 4 that the proportion of non-flowing water (P_21_ and P_22_) decreased significantly (*p* < 0.05) compared to NOX-MPs after oxidation. With the increase in the FO concentration, P_22_ first increased and then decreased (*p* < 0.05), where the changing trend of P_23_ is contrary to that of P_22_. The results showed that the low concentration of FOs can convert free water into non-flowing water in the MP gel, which is beneficial to the formation and stability of the gel structure network. Li added 1.0 mg/mL tea polyphenols to oxidized MPs, proving that an appropriate amount of tea polyphenols can reduce the free water content of the gel and protect the structure of MPs [48].

#### 3.4.4. SEM

The effects of FOs on crosslinked MP aggregation are shown in Figure 5. The microstructures of MP gels exposed to the hydroxyl-radical-generating system and different concentrations of FOs were compared to that of the nonoxidized MP gel. A significant difference between protein gels with and without FOs was observed. The nonoxidized MP gel without FOs exhibited a continuous and compact three-dimensional structure. When hydroxyl free radicals oxidized MPs, the compactness of their gel structure was reduced. This is the result of changes in the structure of MP components, which directly affect the gel properties. Additionally, the gel structure became denser and more uniform because the hydrophobic groups were exposed on the surface of MPs, which influenced the formation of disulfide bonds between the sulfide molecules in the amino acids [49]. When 50 μmol/g FOs were added, the (MP gel structure showed lamellar aggregation and formed a dense flake-like microstructure, and some relatively small pores appeared between the micelles, indicating that an appropriate amount of FOs had a positive effect on the formation of protein gels. When 150 μmol/g FOs were added, the MPs slightly agglomerated. The MP gel exhibited a porous and irregular condensation space, suggesting that a higher concentration of FOs cannot form a better protein gel network structure than the control. After adding 200 μmol/g FOs, there was a severe aggregation of MPs. The gel could not form a three-dimensional network with a compact and uniform microstructure, which indicated that high concentrations of FOs promoted further damage to the protein gel. This was due to the combination of oxidized FOs and MPs to form quinone–protein structures, enlarging the gaps within the gel structure. The above results indicate that high concentrations of FOs have an adverse effect on the gel formation of MPs. Similar to the present study, the addition of 6 and 30 μmol/g protein of rutin made the gel structure of oxidized MPs more compact, while the addition of 150 μmol/g protein of rutin resulted in the formation of rutin–MP aggregates with large pore sizes, which destroyed the microstructure of the gel network [50].

### 3.5. Chemical Force of MP Gels

Molecular interactions play a key role in protein function and gel stability, including hydrophobic interactions, hydrogen bonds, ionic bonds, and disulfide bonds. The molecular forces of MP gels are shown in Table 5. The chemical forces maintaining MP gels are mainly hydrophobic interactions and disulfide bonds, maybe owing to the destruction of ionic bonds and hydrogen bonds when they were heating [51]. The contribution of hydrophobic interactions is much greater than that of ion, hydrogen, and disulfide bonds, indicating that hydrophobic interactions are the main chemical forces maintaining the stable conformation of the MP gel. Hydrophobic interactions are directly related to the formation of gels and play an important role in maintaining the tertiary structures of proteins. During the formation of protein gels, FOs can also greatly change the hydrophobic interactions and molecular conformations of proteins, which is necessary for the coinduction of protein gels by FOs. However, when MPs were oxidized by hydroxyl radicals, the gel structure was changed, and the hydrophobic interactions between proteins were reduced.

Disulfide bonds also contributed to the formation of the MP gel and showed an increasing trend with the increase in the added concentration of feruloylated oligosaccharides. Hydroxyl radicals can attack sulfhydryl groups, which are easily oxidized to form disulfide bonds, resulting in an increase in the content of disulfide bonds (*p* < 0.05), and play an important role in MP aggregation and gel formation, which can significantly improve the gel strength. When 50–100 μmol/g FOs were added, the antioxidant properties of FOs could protect the structure from hydroxyl radicals to maintain a good gel structure, and the main chemical forces were still hydrophobic interactions. Iwami et al. [52] speculated that the hydrophobic interactions were the main forces causing salmon MPs to form the three-dimensional network structure when it was heating. However, when 150–200 μmol/g FOs were added, polar groups were introduced, causing a significant decrease in hydrophobic interactions between nonpolar groups (*p* < 0.05). Meanwhile, excess FOs were oxidized by hydroxyl radicals to form quinones, which combined with proteins, resulting in a large number of disulfide bonds formed during the heating process.

The hydrogen and ionic bonds had less of a contribution to the formation and stability of the MP gel compared with hydrophobic interactions and disulfide bonds, and FOs also had little effect on them. In the secondary structures of proteins, there are a lot of hydrogen bonds in the α-helix structure and β-folded structure. FOs can destroy the α-helix structure of the protein and promote the formation of the β-folded structure. This result is consistent with the secondary structure of the MP gel.

## 4. Conclusions

In the current study, the effects of FOs on the physicochemical properties, protein structure, and gel properties of myofibrillar proteins under hydroxyl radical oxidation conditions were investigated. As an antioxidant, FOs can not only protect the side-chain structure of MPs but also maintain the secondary structure of the protein α-helix at low concentrations (50–100 μmol/g protein). In this case, FOs were integrated into MPs, reducing the surface hydrophobicity and enhancing the surface tryptophan fluorescence intensity and thermal stability. Additionally, the rheological elasticity, viscosity, gel strength, and water-holding capacity of the MP gel were also improved, and the running water in the non-MP gel was increased. However, the structure of the MP protein was destroyed with increasing FO concentrations (150–200 μmol/g protein). Crosslinked protein aggregates were observed by SDS-PAGE and appeared as irregular clusters with small pores under SEM. The water-holding capacity also decreased, and the chemical forces between the gels changed from disulfide bonds to hydrophobic forces. The results of this study highlight the need to add adequate levels of antioxidants to meat to prevent adverse effects on the quality of meat products.

## Figures and Tables

**Figure 1 foods-12-01222-f001:**
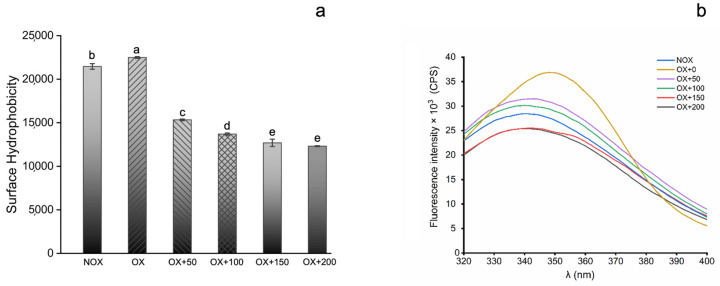
Surface hydrophobicity (**a**) and intrinsic tryptophan fluorescence (**b**) on MPs treated with various concentrations of FOs (0, 50, 100, 150, and 200 µmol/g protein) under nonoxidizing (NOX) or oxidizing (OX) conditions. Different lowercase letters represent significant differences between samples (*p* < 0.05).

**Figure 2 foods-12-01222-f002:**
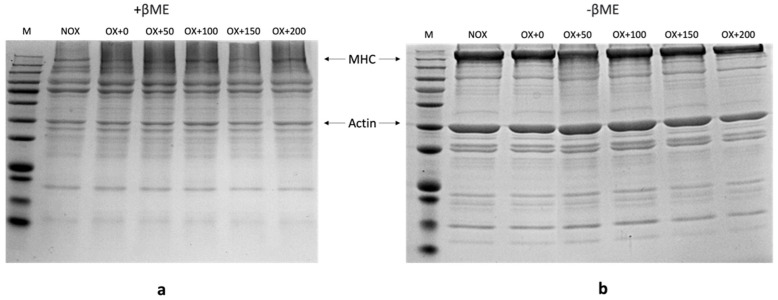
SDS-PAGE patterns of MPs oxidatively stressed in the absence of FOs at different concentrations (0, 50, 100, 150, and 200 µmol/g protein). Samples were prepared in the presence ((**a**), +βMe) or absence of β-mercaptoethanol ((**b**), −βMe). M, marker; NOX, nonoxidized; OX, oxidized; OX + 0, OX + 50, OX + 100, OX + 150, and OX + 200: oxidized in the presence of FOs at 0, 50, 100, 150, and 200 µmol/g protein, respectively.

**Figure 3 foods-12-01222-f003:**
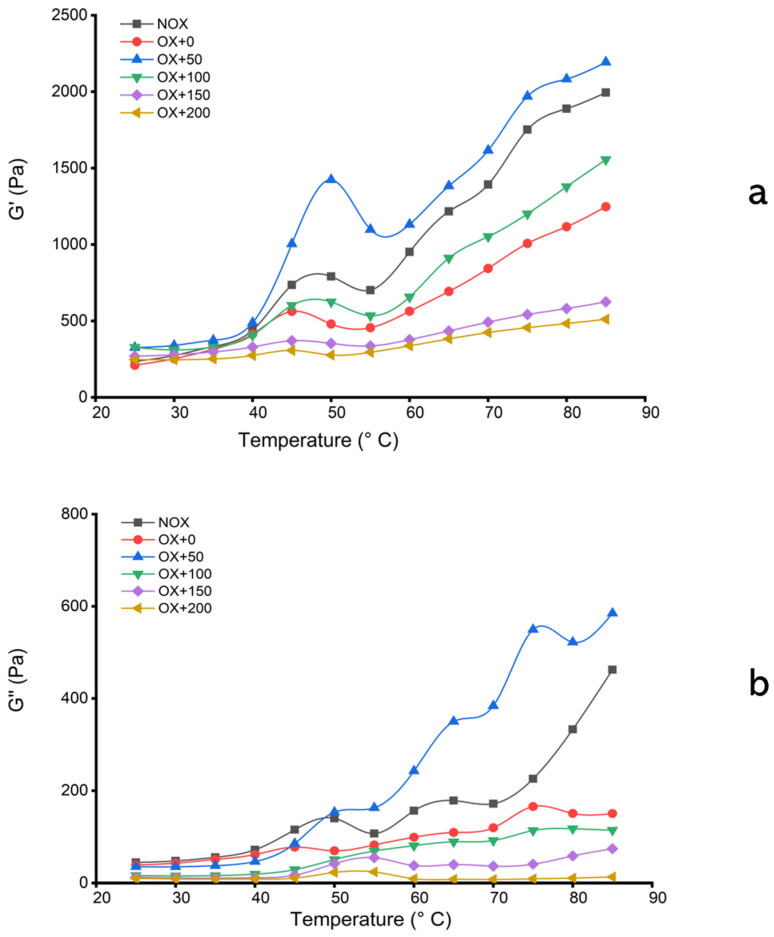
G’ (**a**) and G” (**b**) of MP solutions during gelation (25–85 °C) affected by oxidative stress with or without FOs. NOX, nonoxidized; OX, oxidized; OX + 0, OX + 50, OX + 100, OX + 150, and OX + 200: oxidized in the presence of FOs at 0, 50, 100, 150, and 200 µmol/g protein, respectively.

**Figure 4 foods-12-01222-f004:**
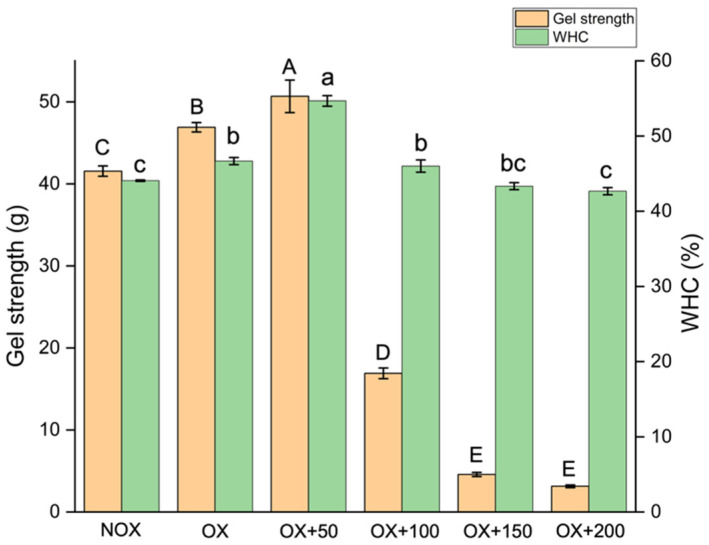
Gel strength and water-holding capacity (WHC) of MP gel treated with various concentrations of FOs (0, 50, 100, 150, and 200 µmol/g protein) under nonoxidizing (NOX) or oxidizing (OX) conditions. Lowercase and uppercase letters represent significant differences in the WHC and gel strength indices between samples (*p* < 0.05).

**Figure 5 foods-12-01222-f005:**
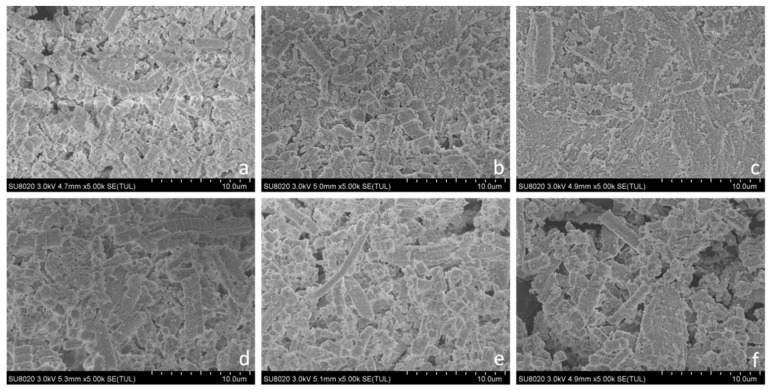
SEM photographs (5000×) of MP gels oxidatively stressed in the presence and absence of FOs at different concentrations: (**a**) NOX, nonoxidized; (**b**) 0; (**c**) 50 μmol/g FOs; (**d**) 100 μmol/g FOs; (**e**) 150 μmol/g FOs; and (**f**) 200 μmol/g FOs.

**Table 1 foods-12-01222-t001:** Changes in amino acid side chains in MPs treated with various addition of FOs (0, 50, 100, 150, and 200 μmol/g protein) under nonoxidizing (NOX) or oxidizing (OX) conditions. Values are means ± the standard deviations (SDs). Different lowercase letters indicate significant differences (*p* < 0.05). NOX refers to nonoxidized MPs; OX + 0 refers to oxidized MPs; OX + 50/100/150/200 refers to 50/100/150/200 μmol/g protein FOs added to oxidized MPs, similarly for other tables hereinafter.

Treatments	Carbonyls [μmol/g MP]	Total Sulfhydryls [μmol/g MP]	Free Amines [μmol/g MP]
NOX	2.25 ± 0.32 ^c^	47.63 ± 8.58 ^a^	37.60 ± 0.72 ^a^
OX + 0	5.93 ± 0.32 ^a^	25.87 ± 2.95 ^d^	22.50 ± 0.68 ^d^
OX + 50	2.85 ± 0.49 ^bc^	49.02 ± 0.61 ^a^	33.65 ± 0.71 ^b^
OX + 100	2.34 ± 0.55 ^c^	48.27 ± 5.78 ^a^	37.35 ± 3.60 ^a^
OX + 150	3.13 ± 0.28 ^bc^	44.28 ± 6.11 ^bc^	32.26 ± 0.57 ^b^
OX + 200	3.71 ± 1.04 ^b^	35.01 ± 0.51 ^cd^	30.46 ± 0.60 ^c^

**Table 2 foods-12-01222-t002:** Secondary structures of MPs treated with various concentrations of FOs (0, 50, 100, 150, and 200 μmol/g protein) under nonoxidizing (NOX) or oxidizing (OX) conditions. Different lowercase letters indicate significant differences (*p* < 0.05).

Treatments	α-Helix	β-Sheet	β-Turn	Random Coils
NOX	15.82 ± 1.97 ^b^	37.85 ± 2.64 ^b^	18.88 ± 0.83 ^c^	27.63 ± 3.64 ^a^
OX + 0	13.94 ± 1.73 ^c^	38.05 ± 0.76 ^a^	19.52 ± 0.58 ^b^	27.70 ± 0.29 ^a^
OX + 50	15.63 ± 2.65 ^b^	36.22 ± 2.20 ^c^	19.25 ± 0.73 ^b^	28.74 ± 0.71 ^a^
OX + 100	17.48 ± 2.18 ^a^	35.03 ± 2.62 ^c^	19.77 ± 0.75 ^b^	27.72 ± 1.15 ^a^
OX + 150	15.80 ± 5.94 ^b^	35.52 ± 2.03 ^c^	20.19 ± 2.25 ^a^	28.89 ± 1.88 ^a^
OX + 200	15.37 ± 3.33 ^b^	36.31 ± 2.60 ^c^	19.44 ± 1.74 ^b^	28.44 ± 1.40 ^a^

**Table 3 foods-12-01222-t003:** Thermal ability of MPs prepared with FOs (0, 50, 100, 150, and 200 μmol/g protein) under nonoxidizing (NOX) or oxidizing (OX) conditions analyzed by DSC. Different lowercase letters indicate significant differences (*p* < 0.05).

Treatment	T_max1_ (°C)	T_max2_ (°C)	T_max3_ (°C)	ΔH_1_ (J/g)	ΔH_2_ (J/g)	ΔH_3_ (J/g)
NOX	48.98 ± 0.16 ^b^	56.80 ± 0.18 ^b^	65.31 ± 0.35 ^a^	0.0102 ± 0.0003 ^b^	0.0149 ± 0.0002 ^b^	0.0025 ± 0.0002 ^a^
OX + 0	45.23 ± 0.19 ^c^	55.13 ± 0.39 ^c^	63.68 ± 0.62 ^c^	0.0098 ± 0.0002 ^b^	0.0140 ± 0.0003 ^b^	0.0018 ± 0.0001 ^b^
OX + 50	49.11 ± 0.21 ^a^	57.01 ± 0.18 ^a^	65.01 ± 0.76 ^b^	0.0105 ± 0.0001 ^a^	0.0153 ± 0.0001 ^a^	0.0025 ± 0.0001 ^a^
OX + 100	47.06 ± 0.19 ^b^	54.97 ± 0.39 ^c^	64.88 ± 0.21 ^b^	0.1001 ± 0.0002 ^b^	0.0148 ± 0.0002 ^b^	0.0009 ±0.0001 ^c^
OX + 150	44.91 ± 0.35 ^c^	54.71 ± 0.43 ^c^	-	0.0087 ± 0.0001 ^c^	0.0115 ± 0.0001 ^c^	-
OX + 200	44.78 ± 0.21 ^c^	-	-	0.0083 ± 0.0002 ^c^	-	-

**Table 4 foods-12-01222-t004:** Effect of oxidation and different concentrations of FOs (0, 50, 100, 150, and 200 μmol/g protein) on T_2_ relaxation time and percentage of T_2_ relaxation time peak area of MP gel. Different lowercase letters indicate significant differences (*p* < 0.05).

Treatments	T2b (ms)	T_21_ (ms)	T_22_ (ms)	T_23_ (ms)	P_2b_ (%)	P_21_ (%)	P_22_ (%)	P_23_ (%)
NOX	0.32 ± 0.07 ^a^	30.78 ± 4.23 ^a^	204.33 ± 2.99 ^a^	527.5 ± 0.03 ^d^	0.85 ± 0.05 ^a^	0.39 ± 0.04 ^e^	75.82 ± 0.06 ^c^	22.91 ± 0.09 ^b^
OX + 0	0.37 ± 0.05 ^a^	19.34 ± 1.43 ^b^	174.58 ± 2.47 ^b^	1096.08 ± 23.27 ^c^	0.90 ± 0.07 ^a^	0.27 ± 0.03 ^e^	63.30 ± 0.17 ^d^	35.53 ± 0.19 ^a^
OX + 50	0.17 ± 0.03 ^a^	15.88 ± 0.90 ^c^	156.17 ± 5.65 ^c^	1844.27 ± 32.57 ^a^	0.88 ± 0.06 ^a^	0.59 ± 0.05 ^a^	87.83 ± 0.02 ^a^	10.70 ± 0.17 ^d^
OX + 100	0.15 ± 0.05 ^a^	12.37 ± 1.31 ^d^	131.97 ± 0.76 ^d^	1672.50 ± 49.52 ^b^	0.87 ± 0.04 ^a^	0.57 ± 0.04 ^b^	89.39 ± 0.05 ^a^	9.17 ± 0.02 ^d^
OX + 150	0.10 ± 0.04 ^a^	10.58 ± 0.99 ^d^	86.20 ± 0.10 ^e^	1418.71 ± 96.01 ^c^	0.85 ± 0.07 ^a^	0.57 ± 0.02 ^b^	88.73 ± 0.02 ^a^	9.85 ± 0.48 ^d^
OX + 200	0.14 ± 0.04 ^a^	27.65 ± 2.22 ^a^	171.30 ± 4.90 ^b^	1283.58 ± 283.13 ^c^	0.86 ± 0.05 ^a^	0.48 ± 0.03 ^c^	80.91 ± 0.27 ^b^	17.75 ± 0.02 ^c^

**Table 5 foods-12-01222-t005:** Chemical forces of MP gels treated with various concentrations of FOs (0, 50, 100, 150, and 200 μmol/g protein) under nonoxidizing (NOX) or oxidizing (OX) conditions. Different lowercase letters indicate significant differences (*p* < 0.05).

Treatments	Ionic Bonds	Hydrogen Bonds	Hydrophobic Interactions	Disulfide Bonds
NOX	0.06 ± 0.02 ^a^	0.08 ± 0.04 ^a^	0.57 ± 0.08 ^a^	0.59 ± 0.09 ^b^
OX + 0	0.05 ± 0.02 ^a^	0.07 ± 0.02 ^a^	0.43 ± 0.02 ^c^	0.68 ± 0.09 ^a^
OX + 50	0.04 ± 0.02 ^a^	0.09 ± 0.02 ^a^	0.55 ± 0.06 ^a^	0.55 ± 0.08 ^bc^
OX + 100	0.06 ± 0.01 ^a^	0.09 ± 0.02 ^a^	0.57 ± 0.12 ^a^	0.50 ± 0.06 ^bc^
OX + 150	0.05 ± 0.01 ^a^	0.10 ± 0.03 ^a^	0.42 ± 0.02 ^b^	0.59 ± 0.04 ^b^
OX + 200	0.04 ± 0.01 ^a^	0.10 ± 0.02 ^a^	0.40 ± 0.04 ^b^	0.64 ± 0.04 ^ab^

## Data Availability

The data presented in this study are available on request from the corresponding author.

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
