# Peer review of "Insights into Feruloylated Oligosaccharide Impact on Gel Properties of Oxidized Myofibrillar Proteins Based on the Changes in Their Spatial Structure"

_foods, 2023, doi:10.3390/foods12061222_

Round 1

Reviewer 1 Report

I read this paper and I didn't identify any scientific novelty. The authors put significant effort that I acknowledge, but it is a work strongly similar to several available papers in the field. The electrophoresis is very basic, and it is not giving any insights, due to the lack of use of any densitometry analyses. Further, two-dimensional analyses are more appropriate for such objective.

Author Response

Dear Reviewer:

We are very grateful to Reviewer for reviewing the paper so carefully.

We have carefully considered the suggestion of Reviewer and made some changes.

Best regards,

Bing Zhao

Reviewer 2 Report

Comments to Authors

 Article “Insights into Feruloylated Oligosaccharides impact on Gel Properties of Oxidized Myofibrillar Proteins based on the changes in their spatial structure highlights the effect of oxidation on proteins. Different points must be considered.

1. Abstract part needs to rewrite, in abstract part add about objective, main results and general conclusion. Also add the numerical values in this portion.

2. Justify the objective of study in the Introduction section and in introduction section add main content only. Also explain when other sources are available for prevent the oxidation then why oxidation is selected.

3. Also explore undesirable effects of oxidation on meat quality.

4. In method of rheological properties add the concentration of sample used for measurement.

5. Below tables, explain the full name of samples to avoid any confusion.

6. In discussion portion add the study of other researchers to compare results, so that betterment of this study may be declared.

7. Gel strength and water-holding capacity of protein decreased drastically, what will be the effect of this on its meat quality.

8. Conclusion portion needs to rewrites.

Author Response

(The authors gave the same response as above.)

Reviewer 3 Report

In the present study, the authors study the impact of Feruloyl oligosaccharide on inhibited oxidation and gel formation of myofibrillar proteins based on the changes in their spatial structure.  This is a very interesting paper. Furthermore, this work in present form presents some imperfections according to the following comments.

Comments to Authors:

 -Page 2; line 8: “in vivo and in vitro” and in all in the paper, should be written in italic

- Page 7; table1; table 2; table 3; table 4and table 5: the meaning of the indices a,b,c and d marked on the data must be given.

- Page 9; line 393: “3.2.4 DSC” the title must not be an abbreviation

- Page 13; figure 4: it is necessary to give the meaning of the lowercase and uppercase letters marked on the histograms.

- Page 14; line 553: correct the title of table 4.

Author Response

(The authors gave the same response as above.)

Reviewer 4 Report

The manuscript “Insights into Feruloylated Oligosaccharides impact on Gel Properties of Oxidized Myofibrillar Proteins based on the changes in their spatial structure” is generally very well written and contains data of some relevance for a general readers as well as of high relevance for specialists in the topic. Although the subject of the paper could be of interest for the readers of the journal, the paper needs some corrections.

Strengths of the paper:

Comprehensive research is very well planned and conducted. The methods are described in detail. Many research methods were used. Conclusions are concisely formulated and follow logically from research observations. A review of the literature is sufficient.

Weaknesses of the paper:

In my opinion, the discussion of the results should be more extensive

Lines: 104  and 114 -  subscripts must be used in the chemical formula (Na2HPO4, FeCl3).

Line 133: I think the abbreviation “SDS” should be clarified.

Line 289: Instead of the word "had" it is better to use the word "was characterized by".

Table 3: I suggest that you include sample DSC curves in the manuscript or in the additional materials.

Author Response

(The authors gave the same response as above.)

Round 2

Reviewer 1 Report

My comments for this paper remain unchanged. There is a lack of novelty in this work.